# Incidence and predictors of acute kidney injury among adults admitted to the medical intensive care unit of a Comprehensive Specialized Hospital in Central Ethiopia

Taye Mezgebu Ashine[1]*, Migbar Sibhat Mekonnen[2], Asnakech Zekiwos Heliso[1], Yesuneh Dejene Wolde[3], Getachew Ossabo Babore[1], Zerihun Demisse Bushen[4], Elias Ezo Ereta[1], Sentayehu Admasu Saliya[1], Bethelhem Birhanu Muluneh[4], Samrawit Ali Jemal[3]

1 Department of Nursing, College of Medicine and Health Science, Wachemo University, Hosanna, Ethiopia, 2 Department of Pediatric and Child Health Nursing, College of Medicine and Health Science, Dilla University, Dilla, Ethiopia, 3 Department of Midwifery, College of Medicine and Health Science, Wachemo University, Hosanna, Ethiopia, 4 Department of Pediatric and Child Health Nursing, College of Medicine and Health Science, Wachemo University, Hosanna, Ethiopia

* tayemezgebu26@gmail.com

**Data Availability Statement:** All relevant data are within the paper and its Supporting Information files.

## Abstract

### Background

Acute kidney injury is a prevalent complication in the Intensive Care Unit (ICU) and a significant global public health concern. It affects approximately 13 million individuals and contributes to nearly two million deaths worldwide. Acute kidney injury among Intensive Care Unit patients is closely associated with higher rates of morbidity and mortality. This study aims to assess the incidence of acute kidney injury and identify predictors among adult patients admitted to the medical Intensive Care Unit.

### Method

A retrospective follow-up study was conducted by reviewing charts of 317 systematically selected patients admitted to the Intensive Care Unit from September 1, 2018, to August 30, 2022, in Wachemo University Nigist Ellen Mohammed Memorial Comprehensive Specialized Hospital. The extraction tool was used for the data collection, Epi-data version 4.6.0 for data entry, and STATA version 14 for data cleaning and analysis. The Kaplan-Meier, log-rank test, and life table were used to describe the data. The Cox proportional hazard regression model was used for analysis.

### Results

Among the total study participants, 128 (40.4%) developed Acute Kidney Injury (AKI). The incidence rate of Acute Kidney Injury was 30.1 (95% CI: 25.33, 35.8) per 1000 person-days of observation, with a median survival time of 23 days. It was found that patients with invasive mechanical ventilation (AHR = 2.64; 95% CI: 1.46–4.78), negative fluid balance (AHR =

**Funding:** This research received funding from Wachemo University since the project was the winner of the 2022 academic year grant.

**Competing interests:** The authors declared no conflict of interest.

2.00; 95% CI: 1.30–3.03), hypertension (AHR = 1.6; 95% CI: 1.05–2.38), and a vasopressor (AHR = 1.72; 95% CI: 1.10–2.63) were independent predictors of acute kidney injury.

## Conclusion

The incidence of Acute Kidney Injury was a major concern in the ICU of the study area. In the intensive care unit (ICU), it was found that patients with vasopressors, invasive mechanical ventilation, negative fluid balance, and chronic hypertension were independent predictors of developing AKI. It would be better if clinicians in the ICU provided targeted interventions through close monitoring and evaluation of those patients with invasive ventilation, chronic hypertension, negative fluid balance, and vasopressors.

## Introduction

Acute kidney injury (AKI) is characterized by an abrupt decrease in kidney function that occurs within hours and encompasses both structural damage and loss of function [1]. The glomerular filtration rate is commonly used as the best overall index of kidney function in critically ill patients [2, 3]. Acute kidney injury is a common problem in both adult and pediatric populations associated with increased morbidity and mortality [4, 5]. AKI is a common complication in the Intensive Care Unit [6, 7]. It is one of the most common risk factors for increased mortality in the ICU [8].

Globally, it is a major public health concern and can be variable between high- and low-income nations. It affects more than 13 million population, and results in 1.7 million deaths each year worldwide [9, 10]. It is a serious life-threatening disease and a global burden in both underdeveloped and developed countries. A study conducted in the USA found that 1.2 million people per year develop AKI during the hospital stay, of which 300,000 people die annually [11]. Moreover, studies showed that the prevalence of AKI IN the ICU is extremely variable between 1% to 66% [4]. Various published studies revealed that high incidence of AKI in the intensive care unit, in South Africa at 58.5% [12], Tanzania at 55.3% [13], Sudan at 39.0% [14], Egypt at 37.4% [15], China 51.0% [16], Norway 38% [17], Brazil 40.3% [18], and Jordan 31.6% [19]. Once developed, AKI is associated with increased death rates amongst critically sick patients, prolongs hospital stays, and is highly expensive to manage clinical costs [20].

Acute Kidney Injury (AKI) is a major problem in Ethiopia, causing high rates of both mortality and morbidity. Studies conducted in Ethiopia have revealed varying death rates for individuals with acute kidney injury (AKI). One research study focused on hemodialysis patients in Ethiopia, and it reported that 17% of the patients died by the end of the follow-up period [21]. Another study conducted in Addis Ababa reported a mortality rate of 33.8% for acute renal failure [22]. Additionally, a separate study indicated that 12.8% of AKI cases resulted in in-hospital mortality [23]. It is widely known that Acute Kidney Injury (AKI) is associated with increased death rates in intensive care unit (ICU) settings worldwide. However, there is limited data in Ethiopia regarding the incidence, risk factors, and outcome of AKI specifically in ICU settings.

Several reviewed kinds of literature demonstrated multifactorial risk factors associated with AKI, and the aetiology and risk factors for acute kidney injury (AKI) vary geographically. These include acute tubular necrosis, Socio-demographic Characteristics, environmental factors, dehydration, trauma, and sepsis from fulminant infections, which can include malaria, tuberculosis, gastroenteritis, diabetic mellitus, hypertension, obesity, cardiovascular diseases,

hypovolemia, and opportunistic infections common in HIV/AIDS patients, among other conditions [15, 24–28].

Acute Kidney Injury is a common issue everywhere that arises in a variety of contexts and hurts patient outcomes, particularly for ICU patients. In developing countries like Ethiopia, data on the incidence, risk factors, and prognosis of patients hospitalized in critical care units is limited. The government's focus on AKI in the ICU involves addressing disparities between countries, leveraging clinical expertise for prevention, and emphasizing the need for further research to enhance patient care and outcomes. Conclusively, by doing more research on Clinical Practice, and risk factor knowledge, the research gaps may be filled and the management and outcome of AKI acquired in the intensive care unit greatly improved. The disease needs to be better recognized in the area due to the lack of scientific evidence on the risk factors, aetiology, incidence rate, and result of AKI. As a result, it diminishes government visibility and hampers efforts to provide quality care in the intensive care unit. According to our knowledge, we have found that limited studies have been conducted in Ethiopia, particularly in the study area. In this retrospective study, we investigated the incidence and predictors of AKI in critically ill adult patients who were admitted to the medical ICU of Wachemo University Nigist Ellen Mohammed Memorial Comprehensive Specialized Hospital.

## Methods

### Study area and period

The study was conducted at Wachemo University Nigist Ellen Mohammed Memorial Comprehensive Specialized Hospital, which is located in the Central Ethiopia Region of Ethiopia. It is one of the teaching hospitals in Ethiopia. It is found in Hosanna, which is the capital city of the region, 230 km away from Addis Ababa, and the capital city of Ethiopia. The hospital provides services for more than 5 million people, both outpatients and inpatients, including emergency and intensive care units. The hospital's adult medical ICU consists of eight beds, each equipped with a heart monitor, perfusor, and mechanical ventilator. The medical ICUs are staffed by a team of healthcare professionals, including ten nurses, two anesthesiologists, two critical care nurse practitioners, one respiratory therapist, and one emergency medicine and critical care physician. This demonstrates the critical care capabilities of the central Ethiopia region. The study period was from March 1 to 30, 2023, by reviewing four years of follow-up data on charts of medical ICU patients who were admitted from September 1, 2018, to August 30, 2022, GC.

### Study design

An institutional-based retrospective follow-up study was conducted among adult patients who were admitted to the medical intensive care unit of Wachemo University Nigist Ellen Mohammed Memorial Comprehensive Specialized Hospital.

### Source of population

All adult patients who were admitted to the medical Intensive Care Unit of Wachemo University Nigist Ellen Mohammed Memorial Comprehensive Specialized Hospital.

### Study population

All adult patients who were admitted to the Medical Intensive Care Unit (ICU) of Wachemo University Nigist Ellen Mohammed Memorial Comprehensive Specialized Hospital during the study period.

### Eligibility criteria

All adult patients whose age is greater than or equal to 18 years, admitted to ICU for more than 24 hours from September 1, 2018, to August 30, 2022, at Wachemo University Nigist Ellen Mohammed Memorial Comprehensive Specialized Hospital, and whose chart is complete and available during data collection period were included in the study whereas, patient with a previous history of dialysis, renal transplant, or admission with end-stage renal disease. Patients' records with no baseline records (serum creatinine) were excluded from the study.

### Sample size determination and sampling procedure

The sample size was determined by Using Stata statistical software version 14 by applying Cox model assumptions considering Cardiovascular disease as a predictor variable from a previous study with an effect size of (hazard ratio) of 1.6 and probability of event 0.4590 [29]. Finally, the required sample size was estimated to be 317. Hence, 317 charts were retrieved. A systematic sampling technique was applied to select charts to be reviewed, including every two charts after determining the first case by lottery methods. The subsequent charts were considered if it appeared that any charts had been missed or had incomplete data.

### Operational definition and measurements

**Acute Kidney Injury(AKI)**: Acute Kidney Injury is defined as if one of the following criteria is met; an increase in serum creatinine level by 0.3 mg/dL or more within 48 hours or an Increase in serum creatinine to 1.5 times baseline or more within the last 7 days or Urine output less than 0.5 mL/kg/h for 6 hours using Kidney Disease Improving Global Outcomes (KDIGO) criteria [3]. **The event was** considered when acute kidney injury occurred on follow-up after 24 hours of Intensive Care Unit admission to the end of the study while those patients who were discharged without AKI, remained on following during the study period, transferred out, and died without AKI due to other causes during the follow-up period were considered as **censored. Survival time** was the time measured in days from the admission of ICU to the development of Acute Kidney Injury or censored. **Baseline period,** which was starting after 24 hours of ICU admission. The **triage score** at admission was determined using the **A**irway, **B**reathing **C**irculation and **D**isability (ABCD) triage score system, which was categorized into three categories: mild (green) if the patient scored between 0 and 4, moderate (yellow) if the patient scored 5–8, and severe (red) if the patient scored greater than 8. For every patient in the critical care unit, we calculated the **cumulative fluid balance** using the percentage (%) from the date of ICU admission until the end of the follow-up (event or censored). Therefore, using the daily cumulative fluid balance (%) subtraction daily input to output in liters, divided by the patient's admission weight and multiplied by 100 for every ICU patient, we classified the patient's fluid balance as positive if the patient recorded a cumulative fluid balance of ≥0% and negative fluid balance if the patient recorded a fluid balance of < 0% [30].

 **Intensive Care Unit Complications** were considered if one or more of the following clinical conditions were diagnosed during follow-up that were not present upon admission: hospital-acquired infection, thromboembolism, shock, acute lung injury, acute liver injury, electrolyte imbalance, delirium, and arrhythmia. **Comorbidity** in this study refers to the presence of one or more pre-existing conditions such as congestive heart failure, hypertension, diabetes mellitus, malignancy, stroke, chronic pulmonary disease, bronchial asthma, or HIV/AIDS during admission.

## Data collection and procedure

First, the information on the patient charts was reviewed and an appropriate data extraction checklist was prepared in English language. The checklist was adapted from intensive care unit patient monitoring, and triage sheets, and by reviewing different related articles [7, 12, 20, 27, 31]. The tool comprised socio-demographic details, the date of the patient's admission to the intensive care unit (ICU), the diagnosis, the baseline vital signs, the baseline laboratory results, comorbidities, the medication administered, the vital signs at the end of the follow-up, the laboratory results at the end of the follow-up, complications, the date of the last follow-up (the date of the developing AKI, patient's recovery or death from the ICU). The list of charts was obtained from the database in the electronic recording systems and ICU registration logbooks using the patient's medical record numbers (MRNs). Three data collectors and one supervisor were employed, and data collection was accomplished within a month from March 1 to 30, 2023.

## Data quality control

The checklist was pretested on 16(5% of the participants) in randomly selected patient charts to keep data quality before starting the actual data collection period, which were not included in the final analysis. Before data collection began, two days of training about the procedure and tool were provided to data collectors and a supervisor. The data collection procedure was monitored closely by the supervisor and principal investigators daily for completeness and consistency of collected data.

## Data processing and analysis

At the end of data collection, the collected data were checked for incompleteness, and then data were entered into Epi-data Manager version 4.6.0.6 and exported to STATA version 14 for cleaning, editing, coding, and analysis. Explanatory data analysis was done to determine missing values, normality tests, and the presence of outliers before data analysis. Then the data were described using relative frequency, percent, mean with standard deviation, and median based on the applicability. A life table was used to estimate the cumulative probability of acute kidney injury at different time intervals. A Kaplan-Meier survival curve was used to estimate median survival time during the follow-up period and log-rank tests were also used to compare survival curves for the presence of differences in the incidence of AKI among the groups.

The Bi-variable analysis was computed to identify possible associations between AKI and each dependent variable using the Cox proportional hazard model. Variables, significant at $p \leq 25\%$ level in the bi-variable analysis, were collectively included in the multi-variable analysis, to identify independent predictors of AKI.

Using variable inflation factors (VIF) between independent variables, multi-co-linearity was checked, and all outputs fell within the acceptable range (mean VIF = 1.27). The proportional hazard assumptions were also checked using the Schoenfeld residuals test (global test, p-value = 0.1575). Besides, model fitness was checked graphically using Cox-Snell residual, which indicates that the model fitted to the data well. In multivariable analysis, statistical significance was declared at $p < 0.05$. The strength of association was reported using an adjusted hazard ratio (AHR) with a 95% confidence interval (95% CI). Finally, data was presented using tables, graphs, and texts.

## Ethical consideration

Ethical consideration was obtained from the institutional review board of Wachemo University (protocol number = WCU-IRB 0011/23). After, approval by IRB, official letters of

cooperation were written for the hospital administrators. The documentation of informed consent was waived by our institutional review board at Wachemo University. Then, data was collected after getting a permission from hospital administrators on behalf of the patients since the study was conducted retrospectively. Data coding aggregates were used to ensure anonymity and confidentiality.

## Result

From the beginning of September 2018 to the end of August 2022, a total of 1058 patients were admitted to the Intensive Care Unit. About 291 charts were excluded using exclusion criteria. Hence, 767 charts were candidates for the review charts, and 317 charts were selected by using a systematic sampling technique to be reviewed, with every two charts included after determining the first case by lottery method. The subsequent charts were considered if it appeared that any charts had been missed or had incomplete data.

### Socio-demographic characteristics

In this study, 184(58.0%) of the study participants were male, of which 76(41.3%) of them developed Acute Kidney Injury [32] whereas 133(39.1%) female participants developed AKI during their ICU stay. The mean age of the study participants was 54.4 years (SD ±18.19356) with a minimum of 19 years and a maximum of 97 years. Regarding the age categories of study participants, the majority of acute kidney injury, 57(46.0%), were recorded among patients after sixty years of age while a small proportion of AKI, 22(29.7%) occurred among those aged < = 40 years (**Fig 1**).

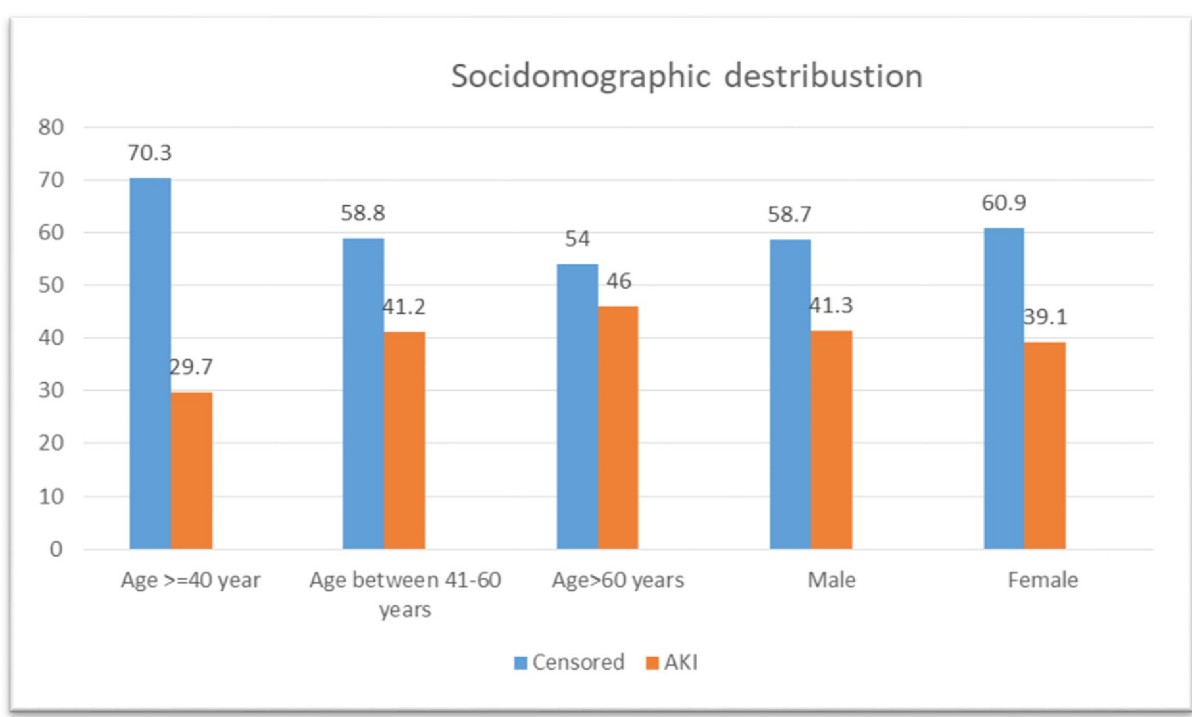

**Fig 1. Socio-demographic characteristics of intensive care unit patients admitted to Wachemo University Nigist Ellen Mohammed Memorial Comprehensive Specialized Hospital, Hosanna, Ethiopia, 2023.**

## Comorbidities and complications of study subjects

Based on the findings, nearly 90% (285/317) of patients were admitted to the Intensive care unit with severe triage scores at admission. The findings showed that 200(63.1%) patients had at least one comorbid condition, of which 46.5% (93/200) had developed AKI. On the contrary, 29.9% (35/117) of the participants who did not have at least one chronic medical illness developed AKI in the ICU. This study's findings revealed that the most prevalent recorded comorbidities were Hypertension 106(33.4%), Diabetics Mellitus 84(26.5%), and Cardiovascular Disease 82(25.9%). The proportion of AKI was much higher among patients having Hypertension (67.9%, n = 106) compared to patients who were free of Hypertension (26.5%, n3 = 211). Furthermore, the study findings revealed that 70.7% of study participants had at least one ICU complication. Hospital Acquired Infections (HAIs) (40.7%, n = 317), Acute Kidney Injury (AKI) (40.4%, n = 317), Shock (31.2%, n = 317), and Delirium (29.3%, n = 317) were the most prevalent recorded ICU complications among study subjects. The study findings also showed that the occurrence of AKI among patients who have ICU complications was (46.9%, n = 224). Conversely, 24.7% (23/93) of the study subjects who did not have any ICU complication developed AKI during their ICU stay (**Table 1**).

## Management and laboratory-related variables

According to the study result, about 54% of the study subjects were intubated, of which (61.45, n = 171) developed AKI, whereas the remaining 46% of the patients were ventilated with Non-invasive ventilation, of which only 15.75% (23/146) of the participants developed AKI. Regarding the baseline vital signs, 174(55%) of the study subjects were found below 90% peripheral oxygen saturation, of which 52.9% (92/174) of the patients developed AKI. Among patients who had recorded an average respiratory rate equal to or greater than 24 breaths per minute, 42.0% (105/250) of the patients developed AKI. Regarding the mean arterial blood pressure of the study subjects, 48.8% (154/3170 of the patients had below 65mmHg, of which 24.7% of the patients developed AKI. Concerning the fluid balance, the majority 252(79.5%) of the study participants had positive fluid balance whereas only 65(20.5%) of the patients had negative fluid balance, of which 60% (39/65) of the study subjects developed AKI. Moreover, the average white blood cell count at baseline was a mean of $12.07 \times 10^3$ c/μl (SD±5.5) with a minimum of $2.3 \times 10^3$ c/μl and a maximum of $31.1 \times 10^3$ c/μl. Almost two- 74.1% (235/317) of the study participants had greater than or equal to 150,000 platelet count at baseline, of which 37.9% of those developed AKI as 47.6% of the patients with below 150,000 c/μl platelet count developed AKI (**Table 2**).

## Description of drug-related variables

This study's findings revealed that all 317(100%) of the study subjects took antibiotics. The proportion of AK among patients who received vancomycin 53.9% (104/193) was higher compared to those who did not receive vancomycin 19.5% (24/124) of patients developed AKI. Among the study participants, only 136(42.9%) of the patients received vasopressors, of which 65.4% (89/136) of the patients developed AKI. In addition, 76.7%, 45.4%, 33.3%, and 28.1% of the patients took thromboprophylaxis, systemic steroids, antihypertensive, and diuretics respectively. Regarding sedation, the majority 60% (189/317) of the study subjects took sedation, of which 56.5% (105/189) of the patients developed AKI (**Table 3**).

## Survival function and incidence rate of acute kidney injury

In this study, the patients were followed for a minimum of 1 day to 48 days, with the median survival time being 23 days (95% CI: 18.0, 33.0). The total person-time observation was 4248

**Table 1. Distribution of clinical characteristics and comorbidities among patients admitted to the intensive care unit of Wachemo University Nigist Eleni Mohammed Memorial Comprehensive Specialized Hospital, Hosanna, Ethiopia, 2023 (n = 317).**

| Variables | Categories | Outcome Status | | |
|---|---|---|---|---|
| | | AKI (%) | Censored (%) | Total (%) |
| Triage Score at Admission | Mild | 4(33.3) | 8(66.7) | 12(3.8) |
| | Moderate | 11(55.0) | 9(45.0) | 20(6.3) |
| | Severe | 113(39.6) | 172(60.4) | 285(89.9) |
| At least one Comorbidities | No | 35(29.9) | 82(70.1) | 117(36.9) |
| | Yes | 93(46.5) | 107(53.5) | 200(63.1) |
| Diabetics Mellitus | No | 87(37.3) | 146(62.7) | 233(73.5) |
| | Yes | 41(48.8) | 43(51.2) | 84(26.5) |
| Hypertension | No | 56(26.5) | 155(73.5) | 211(66.6) |
| | Yes | 72(67.9) | 34(32.1) | 106(33.4) |
| Malignancy | No | 120(39.7) | 182(60.3) | 302(95.3) |
| | Yes | 7(53.7) | 8(46.3) | 15(4.7) |
| Stroke | No | 116(40.3) | 172(59.7) | 288(90.85) |
| | Yes | 12(41.4) | 17 (58.6) | 29(9.15) |
| HIV/AIDS | No | 118(40.8) | 171(59.2) | 289(91.2) |
| | Yes | 10(35.7) | 18(64.3) | 28(8.8) |
| COPD | No | 123(41.1) | 176(59.9) | 299(94.3) |
| | Yes | 5(27.8) | 13(72.2) | 18(5.7) |
| Bronchial Asthma | No | 123(41.7) | 172(58.3) | 295(93.1) |
| | Yes | 5(22.7) | 17(77.3) | 22(6.9) |
| Cardiovascular disease | No | 103(43.8) | 132(56.2) | 235(74.1) |
| | Yes | 25(30.5) | 57(69.5) | 82(25.9) |
| Complication | No | 23(24.7) | 70 (75.3) | 93(29.3) |
| | Yes | 105(46.9) | 119(53.1) | 224(70.7) |
| Thromboembolism | No | 97(39.75) | 147(60.25) | 244(77.0) |
| | Yes | 31(42.5) | 42(57.5) | 73(23.0) |
| Shock | No | 71(32.6) | 147(67.4) | 218(68.8) |
| | Yes | 57(57.6) | 42(42.4 | 99(31.2) |
| Acute lung injuries | No | 107(40.7) | 156(59.3) | 263(83.0) |
| | Yes | 21(38.9) | 33(61.1) | 54(17.0) |
| Acute Liver Injury | No | 111(38.8) | 175(61.2) | 286(90.2) |
| | Yes | 17(54.8) | 14(45.2) | 31(9.8) |
| Hospital Acquired infection | No | 41(21.8) | 147(78.2) | 188(59.3) |
| | Yes | 87(67.4) | 42(32.6) | 129(40.7) |
| Electrolyte Imbalance | No | 90(37.3) | 151(62.7) | 241(76.0) |
| | Yes | 38(50.0) | 38(50.0) | 76(24.0) |
| Delirium | No | 94(42.0) | 130(58.0) | 224(70.7) |
| | Yes | 34(36.6) | 59(63.4) | 93(29.3) |
| Arrhythmia | No | 111(40.5) | 163(59.5) | 274(86.4) |
| | Yes | 17(39.5) | 26(60.5) | 43(13.6) |

Abbreviations: AKI, acute kidney injury; COPD, chronic obstructive pulmonary disease; HIV, human immune Virus; AIDS, acquired immune deficiency syndrome

person-days. The finding of this study revealed that the overall occurrence of AKI among ICU critically ill patients was 128 (40.4%) (95% CI 35.08 to 45.90), and the rest 189 (59.6) of the participants were censored. Among those considered censored (n = 189), 33.4% (106) of the

**Table 2. Distribution of management-related variables among patients admitted to ICU of Wachemo University Nigist Eleni Mohammed Memorial Comprehensive Specialized Hospital, Hosanna, Ethiopia, 2023(n = 317).**

| Variables | Category | Outcome | | |
|---|---|---|---|---|
| | | AKI (%) | Censored (%) | Total (%) |
| Method of oxygen delivery | Non-invasive Ventilation | 23(15.75) | 123(84.25) | 146(46.1) |
| | Invasive ventilation | 105(61.4) | 66(38.6) | 171(53.9) |
| Mode of invasive ventilation | VCV/VAC | 41(60.3) | 27(39.7) | 68(39.8) |
| | PRVC | 32(69.60) | 14(30.4) | 46(26.9) |
| | PCV/PAC | 25(64.10) | 14(35.9) | 39(22.8) |
| | SIMV | 7(38.89) | 11(61.11) | 18(10.5) |
| Noninvasive ventilation | CPAP | 14(13.3) | 91(86.7) | 105(71.9) |
| | NRFM | 9(21.9) | 32(78.1) | 41(28.1) |
| Average Oxygen Saturation | > = 90% | 36(25.2) | 107(74.8) | 143(45.1) |
| | <90% | 92(52.9) | 82(47.1) | 174(54.9) |
| Average Respiratory rate | <24b/m | 23(34.3) | 44(65.7) | 67(21.1) |
| | > = 24b/m | 105(42.0) | 145(58.0) | 250(78.9) |
| MAP baseline | <65mmHg | 38(24.7) | 116(75.3) | 154(48.6) |
| | > = 65mmHg | 90(55.2) | 73(44.8) | 163(51.4) |
| Fluid Balance | Positive | 89(35.5) | 163(64.7) | 252(79.5) |
| | Negative | 39(60.0) | 26(40.0) | 65(20.5) |
| WBC Baseline | < = 11,000 | 57(39.9) | 86(60.1) | 143(45.1) |
| | >11,000 | 71(40.8) | 103(59.2) | 174(54.9) |
| Hemoglobin baseline | <12 mg mg/dl | 25(40.3) | 37(59.7) | 62(19.6) |
| | > = 12mg/dl | 103(40.4) | 152(59.6) | 255(80.4) |
| Platelet baseline | <150,000 | 39(47.6) | 43(52.4) | 82(25.9) |
| | > = 150,000 | 89(37.9) | 146(62.1) | 235(74.1) |
| Creatinine level baseline | <1mg/dl | 94(35.3) | 172(64.7) | 266(83.9) |
| | 1–1.2md/dl | 34(66.7) | 17(33.3) | 51(16.1) |
| BUN at the baseline | <45mg/dl | 99(37.4) | 16662.6) | 265(83.6) |
| | > = 45mg/dl | 29(55.8) | 23(44.2) | 52(16.4) |
| Serum Sodium at the end of follow-up | <135 mEq/l | 45(38.1) | 73(61.9) | 118(37.2) |
| | 135–148 mEq/l | 83(41.7) | 116(58.3) | 199(62.8) |
| Serum Potassium at the end of follow-up | <3.5mEq/l | 10(35.7) | 18(64.3) | 28(8.8) |
| | 3.5-5mEq/l | 48(27.4) | 127(72.6) | 175(55.2) |
| | >5mEq/l | 70(61.4) | 44(38.6) | 114(36.0) |
| WBC at the End of follow-up | < = 11,000 | 24(34.3) | 46(65.7) | 70(22.1) |
| | >11,000 | 104(42.1) | 143(57.9) | 247(77.9) |

**Abbreviations:** AKI, acute kidney injury; BUN, blood urea nitrogen; CPAP, continuous positive airway pressure; MAP, mean arterial pressure; Mg, mill gram; NRFM, non-reservoir face mask; PAC, pressure assist control; PCV, pressure control ventilation, PRVC, pressure regulated volume control; SIMV, synchronized intermittent mandatory ventilation; VCV, volume control ventilation; WBC, white blood cells

patients died, 22.4% (71) of patients were discharged alive, and the rest 3.8% (12) of the patients transferred to other institutions. The incidence rate of Acute Kidney injury among intensive care unit patients was 30.1 (95% CI: 25.33, 35.8) per 1000 person-days observation. The survival probability for AKI at 1, 24, and 48 days were 0.9874, 0.4728, and 0.1359, respectively. The cumulative probability of failure at the end of 12, 24, 36, and 48 days were 0.3582, 0.5272, 0.6612 and 0.8641 respectively (**Table 4**).

**Table 3. Distribution of drug management of patients admitted to the intensive care unit of Wachemo University Nigist Eleni Mohammed Memorial Comprehensive Specialized Hospital, Hosanna, Ethiopia, 2023(n = 317).**

| Variables | Categories | Outcome Status | | |
|---|---|---|---|---|
| | | AKI | Censored | Total (%) |
| Ceftazidime | No | 68(40.2) | 101(59.8) | 169(53.3) |
| | Yes | 60(40.5) | 88(59.5) | 148(46.7) |
| Vancomycin | No | 24(19.35) | 100(80.65) | 124(39.1) |
| | Yes | 104(53.9) | 89(46.1) | 193(60.9) |
| Meropenem | No | 83(35.9) | 148(64.1) | 231(72.9) |
| | Yes | 45(52.3) | 41(47.7) | 86(27.1) |
| Gentamycin | No | 123(40.2) | 183(59.8) | 306(96.5) |
| | Yes | 5(45.45) | 6(54.55) | 11(3.5) |
| Ciprofloxacin | No | 94(43.9) | 120(56.1) | 214(67.5) |
| | Yes | 34(33.0) | 69(67.0) | 103(32.5) |
| Vasopressors | No | 39(26.55) | 142(78.45) | 181(57.1) |
| | Yes | 89(65.4) | 47(34.6) | 136(42.9) |
| Systemic steroid | No | 66(38.2) | 107(61.8) | 173(54.6) |
| | Yes | 62(43.1) | 82(56.9) | 144(45.4) |
| Thrombolytic | No | 28(37.8) | 46(62.2) | 74(23.3) |
| | Yes | 100(41.2) | 143(58.6) | 243(76.7) |
| Antihypertensive | No | 72(34.0) | 140(66.0) | 212(66.9) |
| | Yes | 56(53.3) | 49(46.7) | 105(33.3) |
| Antiviral | No | 109(40.5) | 160(59.5) | 269(84.9) |
| | Yes | 19(39.6) | 29(60.4) | 48(15.1) |
| Diuretics | No | 93(40.8) | 135(59.2) | 228(71.9) |
| | Yes | 35(39.3) | 54(60.7) | 89(28.1) |
| Sedation | No | 23(18.0) | 105(82.0) | 128(40.4) |
| | Yes | 105(55.6) | 84(44.4) | 189(59.6) |
| Ketamine | No | 101(44.3) | 127(55.7) | 228(71.9) |
| | Yes | 27(39.3) | 62(69.7) | 89(28.1) |
| Propofol | No | 115(40.6) | 168(59.4) | 283(89.3) |
| | Yes | 13(38.2) | 21(61.8) | 34(10.7) |
| Ketofol | No | 103(42.0) | 142(58.0) | 245(77.3) |
| | Yes | 25(34.7) | 47(65.3) | 72(22.7) |
| Diazepam | No | 58(28.4) | 146(71.6) | 204(64.35) |
| | Yes | 70(62.0) | 43(38.0) | 113(35.65) |

Abbreviations: AKI, acute kidney injury

**Table 4. Life table of critically ill ICU patients who were admitted at Wachemo University Nigist Eleni Mohammed Memorial Comprehensive Specialized Hospital, Hosanna, Ethiopia, 2023.**

| The time interval, days | Beginning total | Fail | Survival probability | Cumulative failure, probability |
|---|---|---|---|---|
| 0–12 | 317 | 90 | 0.4543 | 0.3582 |
| 12–24 | 144 | 24 | 0.1640 | 0.5272 |
| 24–36 | 52 | 10 | 0.0631 | 0.6612 |
| 36–48 | 21 | 4 | 0.0032 | 0.8641 |

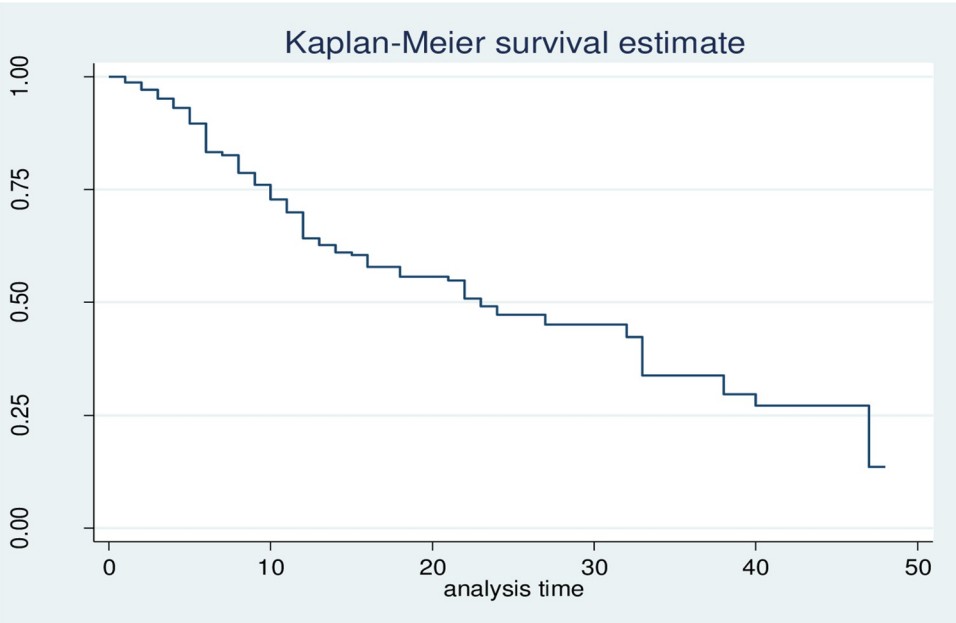

**Fig 2. Overall Kaplan-Meier curve for critically ill ICU patients at Wachemo University Nigist Eleni Mohammad Memorial Comprehensive Specialized Hospital, Hosanna, Ethiopia, 2023(N = 317).** Horizontal axis (X): shows the time of analysis in (days). The vertical axis (Y): indicates survival probability—middle line (downward) survival function.

### Survival status using the Kaplan-Meier curve

The median survival time was 23 days. We can note from the graph that at the initial time of diagnosis the probability of developing AKI was lower, but as follow-up time increased the probability of developing AKI also increased (**Fig 2**).

### Survival function and comparison of survivorship function for the fluid balance category

The Kaplan-Meier estimator survival curve gives the estimate of survival function in the fluid balance category to make a comparison. The survivor function line above another means, that the group defined by the upper line curve had better survival than other group line cures within the category. To test the equality of survival curves in Cochrane; the Mantel Henszel log Rank test was performed. The test statistics obtained from the Log Rank showed there is a statistical difference to test the null hypothesis, which shows there is a difference in the distribution of survival time among categorical variables (Table 5).

In this study, the negative fluid balance has a higher probability of developing AKI with a median time of 12 days (95% CI; 10–27) as compared with positive fluid balance. At the end of the follow-up, the probability of developing AKI of negative and positive fluid balance was found to be 16.3% and 14.6% respectively. The difference was statistically significant with a p-value of 0.0045 (**Fig 3**). Additionally, model fitness was assessed graphically using Cox-Snell residuals, demonstrating that the model fit to the data (**Fig 4**).

### Predictors of acute kidney injury among intensive care unit patients

Cox proportional hazard regression was utilized to determine the predictors of AKI. Bivariable analysis was performed to identify potential variables for confounder adjustment. As a result,

**Table 5. Schoenfeld residual test for proportional assumption of each covariant and overall Cox proportional hazard model among medical intensive care unit patients of Wachemo University Nigist Eleni Mohammed Memorial Comprehensive Specialized Hospital, Hosanna, Ethiopia, 2023.**

| Covariate | Rho | Chi$^2$ | p-value |
|---|---|---|---|
| Comorbidities | 0.11740 | 1.85 | 0.1738 |
| Hypertension | 0.06912 | 0.69 | 0.4054 |
| Method of ventilation | 0.06256 | 0.73 | 0.37445 |
| Respiratory Rate | 0.07677 | 0.93 | 0.3354 |
| Hospital Acquired Infections | 0.04207 | 0.27 | 0.6003 |
| Vancomycin | 0.12121 | 1.98 | 0.1554 |
| Oxygen Saturation | 0.03311 | 0.19 | 0.6623 |
| Average MAP | 0.09844 | 1.18 | 0.2765 |
| Average PIP | 0.03746 | 0.20 | 0.6558 |
| Average PEEP | 0.02865 | 0.14 | 0.7072 |
| Sedation | 0.01734 | 0.05 | 0.8123 |
| Fluid Balance | 0.07681 | 0.82 | 0.3639 |
| Vasopressors | 0.08395 | 0.94 | 0.3323 |
| Global test | | | **0.1575** |

the following variables met the criteria for variable selection: the presence of chronic medical conditions, hypertension, method of ventilation, respiratory rate, hospital-acquired infection, Vancomycin, mean arterial blood pressure, sedation, oxygen saturation level, peak inspiratory pressure, positive end-expiratory pressure, vasopressor, and fluid balance. However, sedation

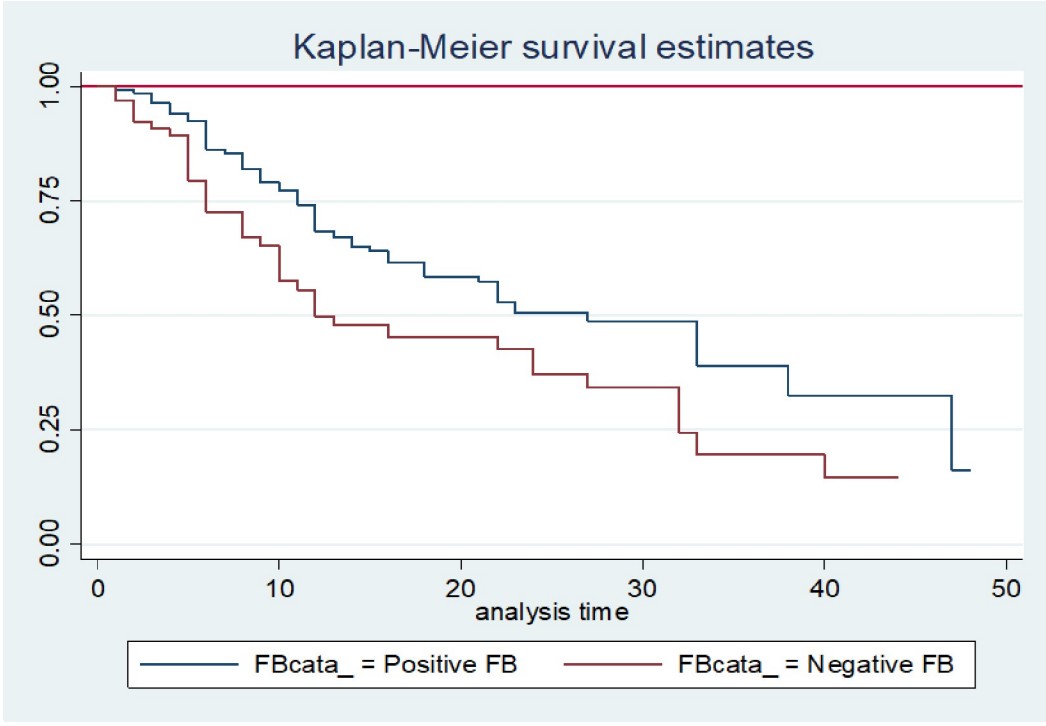

**Fig 3. The Kaplan-Meier curve comparison among fluid balance category for ICU patients at Wachemo University Nigist Eleni Mohammad Memorial Comprehensive Specialized Hospital, Hosanna, Ethiopia, 2023(N = 317).**

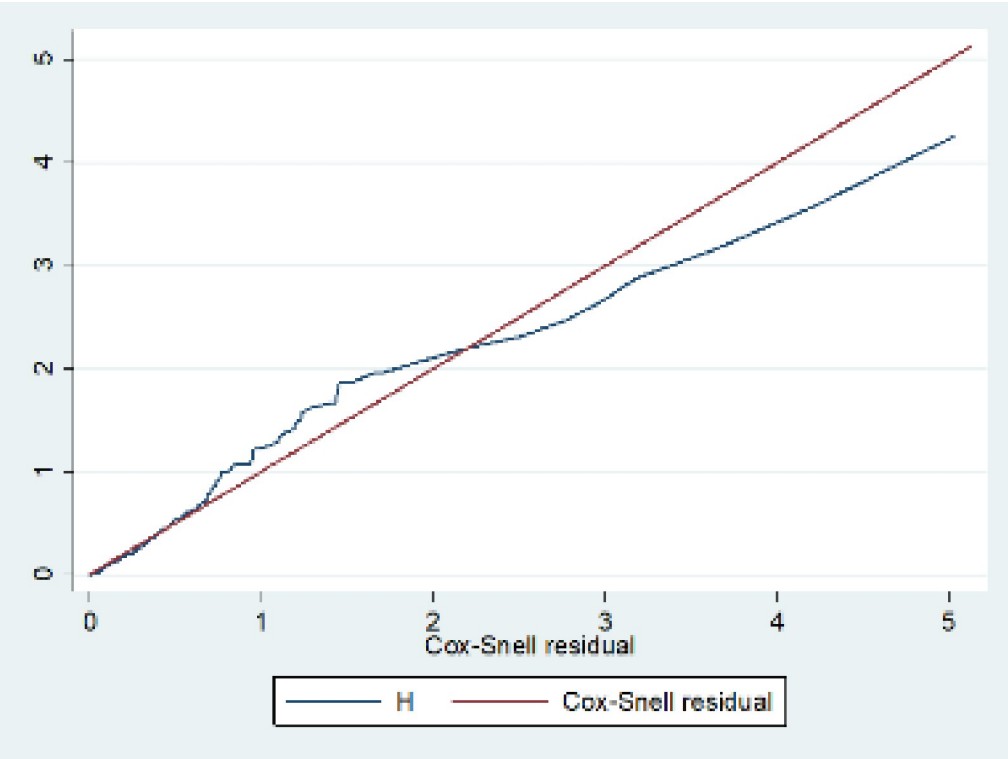

**Fig 4. Displays the Cox-Snell residual test to assess the proportional assumption of each covariate and the overall Cox proportional hazard model.**

and oxygen saturation levels violated the model assumptions and were therefore excluded from the final Cox model.

Afterwards, during multivariable analysis, we found that chronic hypertension, invasive ventilation, negative fluid balance, and vasopressors were statistically significantly associated with AKI at a 95% confidence interval. However, we suspected that fluid balance may have an effect modification, so we conducted a stratified analysis by forming two strata: one with invasive ventilation and one without. The association remained positive, with only a slight change in the hazard ratio. Therefore, the finding is still valid and applicable, even though the presence of effect modification may enhance the magnitude of association due to invasive ventilation. Perhaps, we may need to be cautious when interpreting the magnitude of association (the hazard ratio for fluid balance) (please refer to S1 Table).

Patients having hypertension were 1.60 times increased hazard of developing AKI compared to their counterparts (AHR = 1.6; 95%CI: 1.05–2.38). The hazard of AKI was 2.6 times higher among invasive mechanically ventilated patients compared to patients who ventilated with non-invasive ventilation (AHR = 2.64; 95% CI: 1.46–4.78). On the other hand, the hazard of developing AKI among patients who have negative fluid balance in the ICU was twofold higher compared to their counterparts (AHR = 2.00; 95% CI: 1.30–3.03). Additionally, patients who took vasopressor while in ICU also revealed statistically a significant association with the occurrence of AKI. The hazard of AKI among patients who took vasopressor was 1.7 times (AHR = 1.72; 95%CI: 1.10–2.63) higher than those who did not take vasopressor (**Table 6**).

**Table 6. Bivariate and multivariate analysis of Cox proportional hazard regression among intensive care unit patients admitted in Wachemo University Nigist Eleni Mohammed Memorial Comprehensive Specialized Hospital, Hosanna, Ethiopia, 2023(n = 317).**

| Covariate | Categories | Disease status | | CHR 95%CI | AHR 95% CI | Sig. |
|---|---|---|---|---|---|---|
| | | AKI | Censored | | | |
| Comorbidities | No | 35(29.9) | 82(70.1) | 1 | 1 | |
| | Yes | 93(46.5) | 107(53.5) | 1.62(1.09–2.39) | 0.86(0.561–1.330) | 0.524 |
| Hypertension | No | 56(26.5) | 155(73.5) | 1 | 1 | |
| | Yes | 72(67.9) | 34(32.1) | 3.70 (2.55–5.29) | 1.60(1.052–2.384) | 0.024* |
| Method of ventilation | NIV | 23(15.75) | 123(84.25) | | | |
| | IV | 105(61.4) | 66(38.6) | 5.23(3.31–8.25) | 2.64(1.460–4.783) | 0.002* |
| Respiratory Rate | <24b/m | 23(34.3) | 44(65.7) | | | |
| | > = 24b/m | 105(42.0) | 145(58.0) | 1.75(1.09–2.79) | 1.06(0.6653–1.734) | 0.803 |
| Hospital Acquired Infections | No | 41(21.8) | 147(78.2) | | | |
| | Yes | 87(67.4) | 42(32.6) | 3.95(2.71–7.56) | 1.25(0.772–2.016) | 0.088 |
| Vancomycin | No | 24(19.35) | 100(80.65) | | 1 | |
| | Yes | 104(53.9) | 89(46.1) | 3.94(2.49–6.21) | 1.70(0.983–2.910) | 0.058 |
| Average MAP | > = 65mmHg | 38(24.7) | 116(75.3) | | | |
| | <65mmHg | 90(55.2) | 73(44.8) | 2.51(1.711–3.69) | 1.47(0.981–2.203) | 0.058 |
| Average PIP | <35 cmH2O | 67(59.4) | 132(66.3) | | | |
| | > = 35 cmH2O | 57(59.6) | 39(40.6) | 1.93(1.35–2.77) | 1.14(0.737–1.761 | 0.595 |
| Average PEEP | < = 10cmH2O | 51(26.0) | 145(74.0) | | | |
| | >10cmH2O | 73(75.3) | 24(24.7) | 3.41(2.37–4.90 | 1.54(0.962–2.470) | 0.079 |
| Fluid Balance | Positive | 89(35.5) | 163(64.7) | | | |
| | Negative | 39(60.0) | 26(40.0) | 1.72(1.18–2.51) | 2.00(1.301–3.031) | 0.004* |
| Vasopressors | No | 39(26.55) | 142(78.45) | 1 | | |
| | Yes | 89(65.4) | 47(34.6) | 3.92(2.68–5.74) | 1.72(1.101–2.631) | 0.017* |

**Abbreviation:** AKI, acute kidney injury; AHR, adjusted hazard ratio; CHR, crude hazard ratio; IV, invasive ventilation; NIV, non-invasive ventilation; MAP, mean arterial pressure; PEEP, positive end-expiratory pressure; PIP, peak inspiratory pressure. **Note:** Comorbidities in this study refers to the presence of one or more pre-existing conditions such as congestive heart failure, hypertension, diabetes mellitus, malignancy, stroke, chronic pulmonary disease, bronchial asthma, or HIV/AIDS

## Discussion

This retrospective study aimed to assess the incidence and predictors of Acute Kidney Injury (AKI) among ICU patients. At the end of the follow-up, about 128(40.4%) of the patients developed AKI. The incidence rate of AKI was 30.1 per 1000 person-days of observation. The median survival time was found to be 23 days (95% CI: 18.0, 33.0). Having chronic hypertension, negative fluid balance, invasive mechanically ventilated, and having taken vasopressor during follow-up were found to be independent predictors of developing AKI. The overall incidence of AKI among ICU patients in this study was 40.4%, which is lower than the findings in China, at 51.0% [16], Tanzania at 55.3% [13], and South Africa at 58.5% [12]. This discrepancy may be attributed to variations in study areas, socioeconomic status, and the use of different diagnostic criteria and definitions across the studies. However, the finding of this study was higher than the findings of studies in Sudan, 31.6% [14], Egypt 37.4% [15], Cameron, 22.3% [33], and Australia, 5.5% [34]. The variation in quality care given for ICU, healthcare settings, socio-demographic characteristics, and contextual differences among the countries may account for the differences.

Patients having chronic hypertension were found to be at higher hazard of developing AKI by 1.6 times than non-hypertensive patients while keeping other covariates (AHR = 1.6; 95% CI: 1.05–2.38). This study was compatible with other studies in Sudan [14], Zimbabwe [35],

USA [36], Norway [37], and Ethiopia [38]. The possible justification could be that hypertension in ICU patients under stress results in total peripheral blood vessel resistance, an increase in the amount of extracellular fluid, necrosis of tubular tissue in the nephron, and impaired kidney function, all of which increase creatinine levels, which is the sign of AKI. This is supported by scientific evidence from previous reports [39, 40].

The other factor that was independently associated with the development of AKI among ICU patients was the use of vasopressor during the follow-up period. The patients who took vasopressors during follow–up had a 1.7 times higher hazard of developing AKI than those who did not take vasopressors (AHR = 1.72; 95%CI: 1.10–2.63). This finding is similar to a study conducted in Tanzania [13], and South Africa [12]. The possible justification could be that the use of vasopressors in critically ill patients may potentially lead to decreased renal blood flow and renal oxygen delivery. This severely impaired renal oxygen due to renal vasoconstriction may cause renal ischemia, resulting in Acute Kidney Injury [41, 42]. However, the data on the effect of vasopressor interventions on renal perfusion in critically ill patients are not clear. It needs further investigation of the relationship between vasopressors and renal oxygenation.

Moreover, the negative fluid balance was reported as an independent predictor of AKI among critically ill ICU patients. This finding was contradicted by previous studies conducted in different settings [43, 44]. This discrepancy might be due to the difference in the study population, quality of care given in the ICU and number of settings. However, this finding is supported by previously reported studies [45, 46] that negative fluid balance harms organ function and reduces long-term survival of mortality in ICU. However, future controlled studies to better investigate the relationship between fluid balance and patient outcomes need to take into account the timing of fluid administration, volume, and types of fluid in critically ill patients.

In this study's findings, the hazard of AKI was about 2.6 times higher among invasive mechanically ventilated patients compared to patients who ventilated with non-invasive ventilation (AHR = 2.64; 95% CI: 1.46–4.78). This finding is similar to studies in Brazil [39, 40]. This might be due to the mechanism of invasive mechanical ventilation attributable to Acute Kidney Injury, including hemodynamic instability and selective renal vasoconstriction by mechanical ventilation-induced sympathetic stimulation. Additionally, invasive mechanical ventilation is attributed to the clinical consequences that lead to altered renal hemodynamics [47]. This leads to hypo perfusion to the kidney tissue due to low cardiac output that impaired venous return and leads to ischemic injury that is termed acute renal tubular necrosis [32]. Finally, this leads to a decline in glomeruli filtration rate and elevation of serum creatinine causing the sign of Acute Kidney Injury.

## Limitations of the study

To the best of our knowledge, this study is the first study in Ethiopia to determine the burden of AKI in critically ill ICU patients. This study had its limitations in that it was conducted in a single center using a relatively small sample size. Again, the current study did not evaluate the effect of some variables such as sociodemographic variables, body mass index, serum albumin, and arterial blood gas analysis due to insufficient documentation.

## Conclusion and recommendation

The incidence of AKI was high in the study area and was a major public health concern in the ICU. Patients with hypertension, invasive mechanical ventilation, negative fluid balance, and vasopressors were found to be independent predictors of developing AKI in the intensive care

unit. It would be better if clinicians in the ICU provided targeted interventions through close monitoring and evaluation of those patients on invasive ventilation, chronic hypertension, negative daily fluid balance, and vasopressors. There is limited literature on acute kidney injury (AKI) among intensive care unit (ICU) patients, especially in Ethiopia. Hence, we recommend that further study be needed to state a clear direction on the controversial effect of both negative and positive fluid balance on kidney function. Further, clear data will be needed on ventilation parameters that might be factors for AKI.

### Implication of this study

Because it helps with the early identification of AKI risk factors, this is crucial for patients in acute and life-threatening critical care units when making decisions about patient care. According to this study, AKI is a medical problem that is getting worse in the ICU. Early risk factor management has implications for health policy and the provision of high-quality ICU care, and it may transform practice and enhance patient care generally.

## Supporting information

**S1 Table. Shows that stratified analysis after forming two strata (with invasive ventilation and without invasive ventilation) by keeping the others variables in the multivariate analysis.**
(DOCX)

**S1 File. Data set.**
(DTA)

## Acknowledgments

We would like to acknowledge data collectors, supervisors, staff, and administrators were appreciated for providing the necessary preliminary information. The authors would also like to thank the Wachemo University for giving us this chance.

## Author Contributions

**Conceptualization:** Taye Mezgebu Ashine, Zerihun Demisse Bushen.

**Data curation:** Taye Mezgebu Ashine, Migbar Sibhat Mekonnen, Getachew Ossabo Babore, Elias Ezo Ereta, Sentayehu Admasu Saliya, Samrawit Ali Jemal.

**Formal analysis:** Taye Mezgebu Ashine, Migbar Sibhat Mekonnen, Bethelhem Birhanu Muluneh.

**Funding acquisition:** Taye Mezgebu Ashine, Migbar Sibhat Mekonnen, Asnakech Zekiwos Heliso, Yesuneh Dejene Wolde, Getachew Ossabo Babore, Zerihun Demisse Bushen, Elias Ezo Ereta, Bethelhem Birhanu Muluneh.

**Investigation:** Taye Mezgebu Ashine, Asnakech Zekiwos Heliso, Yesuneh Dejene Wolde, Getachew Ossabo Babore, Zerihun Demisse Bushen, Sentayehu Admasu Saliya, Samrawit Ali Jemal.

**Methodology:** Taye Mezgebu Ashine, Asnakech Zekiwos Heliso, Yesuneh Dejene Wolde, Zerihun Demisse Bushen, Elias Ezo Ereta, Bethelhem Birhanu Muluneh, Samrawit Ali Jemal.

**Project administration:** Taye Mezgebu Ashine, Zerihun Demisse Bushen, Sentayehu Admasu Saliya.

**Resources:** Migbar Sibhat Mekonnen, Asnakech Zekiwos Heliso, Getachew Ossabo Babore, Bethelhem Birhanu Muluneh.

**Software:** Taye Mezgebu Ashine, Elias Ezo Ereta.

**Supervision:** Asnakech Zekiwos Heliso, Yesuneh Dejene Wolde, Getachew Ossabo Babore, Samrawit Ali Jemal.

**Validation:** Taye Mezgebu Ashine, Migbar Sibhat Mekonnen, Yesuneh Dejene Wolde, Getachew Ossabo Babore, Zerihun Demisse Bushen.

**Visualization:** Taye Mezgebu Ashine, Migbar Sibhat Mekonnen, Elias Ezo Ereta, Sentayehu Admasu Saliya, Bethelhem Birhanu Muluneh, Samrawit Ali Jemal.

**Writing – original draft:** Taye Mezgebu Ashine.

**Writing – review & editing:** Taye Mezgebu Ashine, Asnakech Zekiwos Heliso, Yesuneh Dejene Wolde, Sentayehu Admasu Saliya, Samrawit Ali Jemal.

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
