## [Decision Letter · Decision Letter 0]

1 Mar 2024

PONE-D-24-02557Incidence and Predictors of Acute Kidney Injury among Patients in Intensive Care Unit at a Comprehensive Specialized Hospital in Central EthiopiaPLOS ONE

Dear Dr. Ashine,

Thank you for submitting your manuscript to PLOS ONE. After careful consideration, we feel that it has merit but does not fully meet PLOS ONE’s publication criteria as it currently stands. Therefore, we invite you to submit a revised version of the manuscript that addresses the points raised during the review process.

We look forward to receiving your revised manuscript.

Kind regards,

Chiara Lazzeri

Academic Editor

PLOS ONE

Journal Requirements:

"This original research was funded by Wachemio University for data collection and analysis in the 2023 academic year of the university."

"The authors declared no conflict of interest "

5. In the online submission form, you indicated that [Extra data that support the findings of this study are accessible from the corresponding author upon reasonable request and can be shared upon legal request via tayemezgebu26@gmail.com]. 

Reviewers' comments:

Reviewer's Responses to Questions

**Comments to the Author**

1. Is the manuscript technically sound, and do the data support the conclusions?

Reviewer #1: Partly

Reviewer #2: Yes

2. Has the statistical analysis been performed appropriately and rigorously? 

Reviewer #1: Yes

Reviewer #2: Yes

3. Have the authors made all data underlying the findings in their manuscript fully available?

Reviewer #1: Yes

Reviewer #2: Yes

4. Is the manuscript presented in an intelligible fashion and written in standard English?

Reviewer #1: No

Reviewer #2: No

5. Review Comments to the Author

Reviewer #1: 1. Good background, however better to mention the government’s emphasis, standards and the gaps that ought to fill.

2. Was is a retrospective cohort study or follow up study? What is the difference between them?

3. Why patients admitted less than 24 hrs were excluded from the study?

4. “The probability of developing AKI at 1, 24 and 48 days were 0.9874, 0.4728 and 0.1359 consecutively”, it seemed incorrect (coding should be checked) and it seemed survival probability not failure probability.

5. What is the difference between survival time and follow up time? This study reported the median follow up time but not the median survival time, which is important for decision making than the follow up time.

6. Fluid balance: The CHR changed from 1.72 to 2:00 when it is handled in multivariable analysis. What could you think about this effect? (Effect modification)

7. Have you checked the proportional hazard assumption of your model? If so where is the result?

8. Have you checked the model fitness? If so where is the result?

9. Could you think that sample size difference a cause for the discrepancy of results between your study and previous studies? What does mean sampling distribution???

10. All factors identified by this study seems non-modifiable. Could it be used for improving the services, guideline revision and policy improvement?

Reviewer #2: Thank you for the opportunity to read and evaluate the manuscript entitled “Incidence and Predictors of Acute Kidney Injury among Patients in Intensive Care Unit at

a Comprehensive Specialized Hospital in Central Ethiopia: PONE-D-4-02557”.The article covers an interesting topic and suit for publication. However, it needs some revision before being accepted for publication.

General comments:

1. The authors should give line numbers to make the review process easy

2. The manuscript has many editorial problems and needs language editing

Title:

3. The title needs a slight modification to indicate that your study populations are adults admitted to the medical ICU.

Abstract:

4. Avoid/minimize abbreviations: especially don’t start sentences with abbreviations.

5. Mention how many patients developed AKI from the total sample

Introduction:

6. The problem is not well stated. I suggest the authors describe how relevant is AKI as a cause of morbidity and mortality in Ethiopia.

Methods

7. The authors describe that they have reviewed five years of follow-up data on charts of ICU patients who were admitted from September 1, 2018, to August 30, 2022, GC. But it will be 4 years from September 1, 2018, to August 30, 2022.

8. The authors should describe how many beds the medical ICU has, the average annual number of patients admitted to the medical ICU, and the staff composition under the study area section.

9. The source and study population should be corrected. The findings of this study will not be generalized for patients admitted to the surgical ICU and should indicate that the source population for this study will be “all patients admitted to the Medical ICU…… What the authors describe as a study population is the sample included in the study. The study population will be “All patients who were admitted to the medical Intensive Care Unit of Wachemo University Nigist Ellen Mohammed Memorial Comprehensive Specialized Hospital during the study period.”

10. Why do the authors use “admission to the ICU for more than 24 hours” as eligibility criteria? The authors also mention the “Baseline period, which was starting after 24 hours of ICU admission.” Don’t you consider clinical characteristics at the time of admission?

11. Do the authors exclude patients for incompleteness of any study variable?

12. The authors should mention all the parameters used to calculate the sample size and should cite the correct reference where the hazard ratios and probability of event are taken.

13. Why do the authors prefer the systematic sampling technique over the simple random sampling technique?

14. Do the clinicians in the study setting use the given operational definition for AKI? A patient with a serum creatinine level of 0.6mg/dl may come with 0.9 mg/dl within 48 hours of measurement. So, are we going to classify this patient as having AKI? Couldn’t there be any other explanation?

Result:

15. Sociodemographic factors are overlooked. The authors consider only age and sex. Is there any justification for not considering other sociodemographic factors in the study?

16. The authors have a variable “complication” under the comorbidities and complications section. What kind of complication is it? It should be operationalized.

17. I suggest the authors use another table format that doesn’t have d/t color for d/t categories of a variable. Especially, table 2 is not clear.

18. The authors should differentiate between median survival time and median follow-up time. The report is 23 days for both. But I can see from the life table that the survival probability was about 45% around the 12th day. Please re-check the median survival time.

19. I suggest the authors present the table of log-rank test for all important predictor variables rather than showing the comparison of survivorship function for only the fluid balance.

20. The variable comorbidities entered in the multivariable regression should be operationalized.

21. Vancomycin is usually given for hospital-acquired infection but both hospital-acquired infection and vancomycin were considered as separate variables. Do the authors check the correlation b/n these two variables?

22. In Table 5, the sum of the rows is much above the sample size for variable “Average MAP” and below the sample size for variables “Average PIP” and “Average PEEP”. Check the consistency of the sum of rows and columns anyway.

Discussion:

23. The justification given for the differences in the incidence of AKI with other findings should be scientific. Considering sample size, follow-up period, and sampling method as an explanation for the difference doesn’t make sense.

24. The justification for the significant factors seems personal suggestions. The authors should cite scientific evidence for the possible explanations.

25. The authors describe that “this study is the first study in Ethiopia to determine the burden of AKI in critically ill ICU patients.” However, there are some studies in d/t parts of Ethiopia and this is a sign of not acknowledging others effort.

6. PLOS authors have the option to publish the peer review history of their article (what does this mean?). If published, this will include your full peer review and any attached files.

Reviewer #1: No

Reviewer #2: **Yes: **Yaregal Animut (Assistant Professor of Epidemiology and Biostatistics, Department of Epidemiology and Biostatistics, College of Medicine and Health Sciences, University of Gondar, Ethiopia)

---

## [Author Response · Author response to Decision Letter 0]

19 Apr 2024

Manuscript title: Incidence and Predictors of Acute Kidney Injury among Adults Admitted to the Medical Intensive Care Unit of a Comprehensive Specialized Hospital in Central Ethiopia (ID PONE-D-24-02557)

Response to Editor and Reviewers’ Comments 

Dear Editor and reviewers of PLOS ONE journal, 

We the authors of this manuscript would like to thank the editor and reviewers for your thoughtful feedback and comments. We are pleased with your immediate response as well. Here below are the responses to the comments.

Editorial comments 

1. Comment #1: “Please ensure that your manuscript meets PLOS ONE's style requirements, including those for file naming."

Author’s response: We have reviewed the journal styles and requirements and adjusted the files accordingly. 

2. Comment #2: “We note that the grant information you provided in the ‘Funding Information’ and ‘Financial Disclosure’ sections does not match.” 

Author’s response: We have reviewed and tried to amend the required changes to the online submission and the manuscript (Line 483)

Comment #3: “This original research was funded by Wachemio University for data collection and analysis in the 2023 academic year of the university." Please state what role the funders took in the study. If the funders had no role, please state: "The funders had no role in study design, data collection and analysis, decision to publish, or preparation of the manuscript." If this statement is not correct you must amend it as needed. Please include this amended Role of Funder statement in your cover letter; we will change the online submission form on your behalf.”

Author’s response: Thank you! We have corrected it as suggested. 

Comment #4: “Please ensure that your ethics statement is included in your manuscript, as the ethics statement entered into the online submission form will not be published alongside your manuscript.”

Author’s response: Thank you in advance! We have accepted the comments and we have included the ethical statement in the manuscript and rephrased as recommended. 

Reviewer #1: 

Comment#1; “Good background, however better to mention the government’s emphasis, standards and the gaps that ought to fill.”

Author’s responses: We have accepted the comment and put the required amendments to mention the gaps and government emphasis, in the background of the manuscript (kindly see lines 87-91)

Comment#2; “Was it a retrospective cohort or a follow-up study? What is the difference between them?” 

Author’s responses: Actually, retrospective follow-up study, best suits for our study context since we have conducted on single-arm study (all our participants were exposed/admitted to ICU). Thus, we have rephrased it in the main document (Line 316&317). To clarify the difference between the two designs, a retrospective cohort study considers two independent cohorts (considers two-arms). For instance, to apply a cohort study, the participants should have been consisting of patients admitted to the ICU (exposed group) and those not admitted to the ICU (non-exposed group). However, patients who were not admitted to the ICU could not be the right comparison groups to address the objectives of this study. This is because; cohorts should have homogenous characteristics to be considered as comparative groups. That’s why we prefer retrospective follow-up over cohort! 

Comment#3; “Why patients admitted less than 24 hrs were excluded from the study?”

Author’s responses: It is true that we did not consider patients who stayed less than 24 hours in the ICU to maintain the validity and quality of our data and our study findings as well. It was difficult to collect the time variable data in hours since the default time scale in Ethiopian medical registry system is in days, except for some cases such as DKA follow-up, vasopressors, and hypertensive crisis management. Therefore, the length of stay for patients who left the ICU within 24 hours was not clearly documented in hours for most of the registries. In such scenario, we had no any option to handle this issue better than excluding them. Including these cases will affect the incidence of the outcome variable, and we did not want to make this reckless mistake. 

Comment#4; “The probability of developing AKI at 1, 24 and 48 days were 0.9874, 0.4728 and 0.1359 consecutively”, it seemed incorrect (coding should be checked) and it seemed survival probability not failure probability.”

Author’s responses: We appreciate it! That was typing error. We accepted the comment and made the required changes (Please look at lines 264 and 265).

Comment#5; “What is the difference between survival time and follow-up time? This study reported the median follow up time but not the median survival time, which is more important for decision making than the follow-up time.”

Author’s responses: Thank you again, that was a great point! Actually, the finding was deemed to describe the median survival time, and that was mistakenly interpreted as median follow-up time. Thus, we totally agreed with the respected reviewer’s idea and rephrased the term “follow-up time” to "median survival time" in the manuscript (Please see Line 258). 

Comment#6; “Fluid balance: The CHR changed from 1.72 to 2:00 when it is handled in multivariable analysis. What could you think about this effect? (Effect modification)”

Author’s responses: Well, that was a good concern. We were also concerned with this finding, and we run stratified analysis by forming different strata. Accordingly, we noted that the AHR for fluid balance ranges from 2.02-2.26 for each strata with an exception to invasive mechanical ventilation (AHR=1.68) (kindly refer to supplementary file 1). However, the association remains positive (it neither changed the direction of association nor nullified it). Consequently, the finding was still valid and applicable despite the magnitude of association could be enhanced by the effect of another variable since the presence of effect modification could not disqualify the association. Perhaps, we may need to be cautious while interpreting the magnitude of association, and thus, we have described this under the discussion section of the main document (Lines 366-382). We need to remember that the effect of an effect-modifying variable is on the outcome variable, and it could not affect the exposure variable. We could not thank enough the esteemed reviewer though for reminding this essential point. 

Comments#7 “Have you checked the proportional hazard assumption of your model? If so where is the result?” 

Author’s response: Thank you once again! Yes, we have checked the model assumptions using Schoenfeld residuals test for each covariate including the overall global test. The global test result was >0.05 (P-value=0.1575), which indicates the PH-assumptions were fulfilled (kindly refer to line 356-372).

Comment#8; “Have you checked the model fitness? If so where is the result?”

Author’s response: Of course, beside the global test, we have checked the model fitness graphically using Cox-Snell residual graph. The graph runs following the slope of the 45° reference line, which ensures the model applied fitted to the data well. The Cox-Snell graph is presented in the main manuscript file (Please check in Line 220 -226)

Comment#9; “Could you think that sample size difference is a cause for the discrepancy of results between your study and previous studies? What does mean sampling distribution???”

Author’s response: Thank you we accept the comments. We made necessary changes to the discussion part (Kindly, refer to lines 410 and 411). Another issue raised is the uncertainty regarding the inclusion of the sampling distribution in our manuscript. A sampling distribution is a statistical probability distribution calculated using multiple samples taken from a given population. Discrepancies between studies in the current study are not caused by the sample distribution itself. 

Comment 10; “All factors identified by this study seem non-modifiable. Could it be used for improving the services, guideline revision and policy improvement?”

Author’s response: Thank you for your concern. However, we strongly believe that almost all identified factors are modifiable. Some of identified predictors (e.g. hypertension & invasive ventilation) may seem to be unavoidable. However, it does not imply that they are non-modifiable since their effect can be prevented from causing complications such as AKI. For instance, the effect of hypertension can be reduced by focused and cautious screening and management of hypertensive cases at or before admitted to the ICU. Complications of invasive ventilation such as vital organ damage could be reduced using skilled ventilator management and applying patient specific ventilator parameters. On the other hand, the effect some factors like negative fluid balance can be considered as avoidable. Strict fluid balance monitoring (input and output), and adjusting fluid intake of patients could help to acquire neutral fluid balance, so that the effect of negative fluid balance can be minimized. Conversely, this study did not identify factor that are commonly considered as non-modifiable like age and sex.

Reviewer #2: 

General comments:

Comments#1: “The authors should give line numbers to make the review process easy”

Author’s Response: Comment accepted and required changes made.

Comments#2. “The manuscript has many editorial problems and needs language editing”

Authors’ response: We have gone through the entire of manuscript for the grammatical, spelling, punctuation, and editorial errors, and tried to amend accordingly as much as possible. 

Title:

Comment#3. “The title needs a slight modification to indicate that your study populations are adults admitted to the medical ICU.”

Authors’ response: We have moderated the title as directed. (Line 1&2)

Abstract:

Comment#4: “Avoid/minimize abbreviations: especially don’t start sentences with abbreviations.”

Authors’ response: Thank you. We made the required changes (Kindly refer to Lines 34 and 37). 

Comment#5: “Mention how many patients developed AKI from the total sample”

Authors’ response: Thanks again. We incorporated the requested point (Please look at lines 45&46). 

Introduction:

Comment#6. “The problem is not well stated. I suggest the authors describe how relevant is AKI as a cause of morbidity and mortality in Ethiopia.”

Authors’ response: Well, We tried to broaden the problem statement as suggested by the reviewer (Kindly refer to Line 81-90).

Methods

Comment#7: “The authors describe that they have reviewed five years of follow-up data on charts of ICU patients who were admitted from September 1, 2018, to August 30, 2022, GC. But it will be 4 years from September 1, 2018, to August 30, 2022.”

Author’s response: We appreciate it!, It was an edition error and we have rephrased it (Please see line 127). 

Comment#8: The authors should describe how many beds the medical ICU has, the average annual number of patients admitted to the medical ICU, and the staff composition under the study area section.

Author’s response: We admitted the comment and moderated the main document accordingly (kindly look at lines 121-128).

Comment#9: “The source and study population should be corrected. The findings of this study will not be generalized for patients admitted to the surgical ICU and should indicate that the source population for this study will be “all patients admitted to the Medical ICU…… What the authors describe as a study population is the sample included in the study. The study population will be “All patients who were admitted to the medical Intensive Care Unit of Wachemo University Nigist Ellen Mohammed Memorial Comprehensive Specialized Hospital during the study period.”

Author’s response: We have made changes to the source and study population as recommended and tried to amend accordingly as much as possible (please see lines 137-139). 

Comment#10: “Why do the authors use “admission to the ICU for more than 24 hours” as eligibility criteria? The authors also mention the “Baseline period, which was starting after 24 hours of ICU admission.” Don’t you consider clinical characteristics at the time of admission?”

Author’s response: We appreciate your insightful comment! It is true that we did not consider patients who stayed less than 24 hours in the ICU to maintain the validity and quality of our data and our study findings as well. It was difficult to collect the time variable data in hours since the default time scale in Ethiopian medical registry system is in days, except for some cases such as DKA follow-up, vasopressors, and hypertensive crisis management. Therefore, the length of stay for patients who left the ICU within 24 hours was not clearly documented in hours for most of the registries. In such a scenario, we had no option to handle this issue better than excluding them. Including these cases will affect the incidence of the outcome variable, and we did not want to make this reckless mistake.

Comments#11: “Do the authors exclude patients for the incompleteness of any study variable?”

Author’s response: Well, We did not exclude any patient for the incompleteness of any study variable. Rather, we only excluded charts with missing basic variables (e.g. baseline and follow-up serum creatinine level) since it was difficult to determine the outcome status. It was clearly described in the main document (Line 146 &147).

Comments#12: “The authors should mention all the parameters used to calculate the sample size and should cite the correct reference where the hazard ratios and probability of the event are taken.”

Author’s response: Comment admitted and amendments made accordingly (kindly see line 151).

Comments#13: “Why do the authors prefer the systematic sampling technique over the simple random sampling technique?”

Author’s response: Since we used patient charts and registries, data was ordered in time series easy to select by fixed interval from a large number of medical records from the registration book. It also provide the assurance of sampling selection that evenly sampled the population for the true representativeness of the entire documented data on the registration, easy for utilization, increase control and low risk of bias over simple random sampling method. 

Comments#14. “Do the clinicians in the study setting use the given operational definition for AKI? A patient with a serum creatinine level of 0.6mg/dl may come with 0.9 mg/dl within 48 hours of measurement. So, are we going to classify this patient as having AKI? Couldn’t there be any other explanation?”

Author’s Response: Thank you! Yes, of course, the data collectors and supervisors were oriented at the beginning to diagnose the case based on the criteria specified in the operational definition section of the study. Perhaps, they may not use exactly similar diagnostic criteria for routine practice. However, operational definitions were strictly applied for the study during data collection. To determine whether the previous diagnoses on the charts by the physicians were accurate or not, we assessed them against the pre-defined operational definition for this study. If they did not meet the criteria, we excluded the diagnosed AKI by clinicians. 

Result:

Comments#15: “Sociodemographic factors are overlooked. The authors consider only age and sex. Is there any justification for not considering other sociodemographic factors in the study?”

Author’s Response: Yes, it is correct that our sociodemographic data were limited. Unfortunately, we were unable to find detailed records of socio-demographic characteristics such as marital status, occupations, and level of education for ICU patients. This is why we acknowledge the retrospective nature of our study as a limitation. 

Comments#16: “The authors have a variable “complication” under the comorbidities and complications section. What kind of complication is it? It should be operationalized.”

Author’s Response: We accept the comments and made modified the main document as directed (Kindly look at lines 178-183). 4

Comments#17: “I suggest the authors use another table format that doesn’t have d/t color for d/t categories of a variable. Especially, table 2 is not clear.”

Author’s Response: The aut

---

## [Editor Report · Decision Letter 1]

6 May 2024

Incidence and Predictors of Acute Kidney Injury among Adults Admitted to the Medical Intensive Care Unit of a Comprehensive Specialized Hospital in Central Ethiopia

PONE-D-24-02557R1

Dear Dr. Ashine,

We’re pleased to inform you that your manuscript has been judged scientifically suitable for publication and will be formally accepted for publication once it meets all outstanding technical requirements.

Kind regards,

Chiara Lazzeri

Academic Editor

PLOS ONE
---

## [Editor Report · Acceptance letter]

10 May 2024

PONE-D-24-02557R1 

PLOS ONE

Dear Dr. Ashine, 

I'm pleased to inform you that your manuscript has been deemed suitable for publication in PLOS ONE. Congratulations! Your manuscript is now being handed over to our production team.

Kind regards, 

on behalf of

Dr. Chiara Lazzeri 

Academic Editor

PLOS ONE